# An information-theoretic approach to obtain ensemble averages from Earth system models

Carlos A. Sierra[1] and Estefanía Muñoz[1,2]

[1]Max Planck Institute for Biogeochemistry, Jena, Germany
[2]CREAF, Cerdanyola del Vallès, Barcelona, Spain

**Correspondence:** Carlos A. Sierra (csierra@bgc-jena.mpg.de)

**Abstract.** Inferences in Earth system science rely to a large degree on the numerical output of multiple Earth System Models. It has been shown that for many variables of interest, the multi-model ensemble average often compares better with observations than the output from any one individual model. However, a simple arithmetic average does not reward or penalize models according to their ability to predict available observations, and for this reason, a weighted averaging approach would be preferred for those cases in which there is information on model performance. We propose an approach based on concepts from information theory with the aim to approximate the Kullback-Leibler distance between model output and unknown reality, and to assign weights to different models according to their relative likelihood of being the best-performing model in a given grid cell. This article presents the theory and describes the steps necessary for obtaining model weights in a general form, and presents an example for obtaining multi-model averages of carbon fluxes from models participating in the sixth phase of the Coupled Model Intercomparison Project CMIP6. Using this approach, we propose a multi-model ensemble of land-atmosphere carbon exchange that could be used for inferring long-term carbon balances with much reduced uncertainties in comparison to the multi-model arithmetic average.

## 1  Introduction

Inferences in Earth system science (ESS) depend to a large extent on the output from Earth system models (ESMs) that combine different components of the carbon, water, and energy balance of Earth and make spatial and temporal predictions of a large number of variables of scientific interest. These models are relatively large and transcend specific knowledge of single disciplines. Multiple modeling groups run large simulations and participate in coordinated Model Intercomparison Projects (MIP), where the forcings to the model are similar and the outputs across models are comparable (Eyring et al., 2016). For making inferences about a particular aspect of the dynamics of the Earth system, one is confronted with the problem of what model to analyze, how to assign a degree of belief to the results of a certain model, and how to perform averaging across models to obtain an unbiased estimator of some variable of interest (Giorgi and Mearns, 2002, 2003; Hagedorn et al., 2005; Knutti, 2010; Knutti et al., 2010; Hausfather et al., 2022; Tebaldi and Knutti, 2007; Sain and Kleiber, 2025).

These problems are not new to science, and a lot of previous work has been done in the fields of information theory and statistics to address some of these issues. In particular, the problem of model selection and multi-model inference in statistics,

where the idea is to fit multiple models to observed data, is relatively well developed (Anderson, 2007; Burnham and Anderson, 2002; Millington and Perry, 2011; Claeskens and Hjort, 2008). However, classical multi-model inference methods have been developed mostly for simple statistical models, usually expressed as polynomial functions, and not for dynamical models that recursively update a large set of state variables based on a dynamical rule as in ESMs. Therefore, there is a need to expand the existing theory of multi-model inference to the large-dimensional models used in Earth system science.

Three main challenges emerge when trying to expand statistical methods of multi-model inference to ESMs. First, ESMs are not parameterized using statistical techniques such as maximum likelihood estimation (MLE) for the entire set of parameters of the model, which may vary spatially depending on the process being represented. Hirotugu Akaike showed in his seminal work that a measure of distance among models can be obtained with the log-likelihood estimate of the model with respect to an observational set (Akaike, 1974, 1981). Thus, the challenge is to find a way to obtain a distance metric that does not rely

on MLE methods. This is particularly important for ESMs because model uncertainty not only includes parameter uncertainty but also uncertainty due to initial states and boundary conditions (Tebaldi and Knutti, 2007). A second challenge is that the total number of parameters in a given ESM is usually unknown to the user of ESM numerical output. The classical statistical theory assigns penalties to models according to their number of parameters, but this is practically impossible for published output from ESMs because there is no consistent reporting of the type and number of parameters, and whether they were

obtained by an optimization method or agnostically inputted. Thus, the classical theory needs to be modified by an approach that disregards model complexity and does not add a penalty for it. A third challenge is the high-dimensionality of the problem of multi-model inference with ESMs. While the classical theory usually considers one predicted variable and a small set of explanatory variables linked by a polynomial function, the problem of inference in Earth system science is to obtain expected values of a large set of state variables reported in a multi-dimensional lattice (geographical coordinates, time, and height or

depth).

In this article, we propose an adaptation of the classical statistical theory of multi-model inference addressing the challenges of lack of MLE, unknown parameter space, and high dimensionality. We explicitly deal with the problem of model averaging following the conceptual approach described by Burnham and Anderson (2002), which builds on the work developed by H. Akaike in the 1970s and 80s (Parzen et al., 1998).

It has been shown in previous publications that the multi-model average of a variable of interest such as surface air temperature tends to agree better with observations than the predictions of any one single model (Doblas-Reyes et al., 2003; Hagedorn et al., 2005; Elvidge et al., 2023). The arithmetic mean from a set of models gives no consideration about the ability of some models to perform better than others. This is equivalent to assuming that each model is weighted equally in their prediction ability. However, it has been shown that this 'model democracy' is inappropriate for multi-model inference in climate science

(Knutti, 2010; Knutti et al., 2017). Although we are aware that other approaches for multi-model averaging of ESM output have been proposed before (e.g., Knutti et al., 2017; Ribes et al., 2021; Sanderson et al., 2015; Giorgi and Mearns, 2002; Tebaldi and Knutti, 2007; Tebaldi et al., 2005; Merrifield et al., 2020; Elvidge et al., 2023), we are not aware of a previous methodology grounded on information-theoretic principles. In addition, some previous approaches calculate weights as fixed values for all output of a model, but it is desirable to obtain weights based on the ability of some models to provide better estimates for some

spatial regions than others. Our proposed approach operates at the grid-cell level and, therefore, produces weights based on the ability of a model to predict the spatial distribution of a variable of interest.

This article is organized as follows, first, we provide a conceptual derivation of a distance metric appropriate for ESMs using classical concepts from multi-model inference, and a derivation of model weights to obtain ensemble averages and uncertainties. The conceptual framework presented here follows the derivation presented by Burnham and Anderson (2002), but adapted to the specific case of ESMs. Then, we present a step-by-step description of the method applied to multidimensional ESM numerical output. We later apply the method to the computation of the average net carbon exchange between the land and the atmosphere using models from the CMIP6 archive and discuss the results.

## 2 Conceptual approach

We assume that a model $g$ is an approximation of the unknown full reality $f$, and the *divergence* between $g$ and $f$ is given by the Kullback-Liebler information

$$I(f,g) = \int f(x) \log\left(\frac{f(x)}{g(x|\theta)}\right) \mathrm{d}x, \tag{1}$$

where $x$ is the variable being modeled, and $\theta$ is the set of parameters in the model $g$. In this framework, both $f$ and $g$ are understood as probability density functions. In the case of ESM model output, we can think of them as the proportional distribution of some quantity $x$ over a spatial and temporal domain. For ESMs, the vector $\theta$ not only includes the set of parameters in the model, but also the set of initial conditions for the state variables that may lead to different ESM outputs.

The information or divergence $I(f,g)$ is often interpreted as a distance between two distributions even though it does not meet the mathematical definition of a distance metric; i.e. it is not symmetric (the value from $f$ to $g$ can be different than from $g$ to $f$) and it does not satisfy the triangle inequality (Cover and Thomas, 2006). Nevertheless, for the purpose of this article, it is useful to think of $I(f,g)$ as a distance between probability distributions, and we will use the term distance as synonym of divergence throughout this article.

$I(f,g)$ can also be interpreted from the point of view of information theory as the amount of information available to discriminate $g$ from $f$ (Kullback and Leibler, 1951), or as the loss of information from full reality by the model approximation. In principle, we are interested in selecting or ranking models according to $I(f,g)$, but in practice, we cannot compute $f(x)$. This limitation is partly alleviated, as we will see later, by the expansion of equation 1 as

$$I(f,g) = \int f(x) \log f(x)\mathrm{d}x - \int f(x) \log(g(x|\theta))\mathrm{d}x. \tag{2}$$

Notice that each of these terms on the rhs of equation 2 can be considered as an expectation of a probability distribution. Furthermore, because full reality $f$ is a fixed quantity that does not depend on parameters or initial conditions, the first term on the rhs of 2 can be considered as a constant. In terms of expectations, equation 2 can be written as

$$I(f,g) = E_f[\log(f(x))] - E_f[\log(g(x|\theta))]. \tag{3}$$

This equation suggests that for a given model $g$, there would be a set $\theta_m$ that minimizes the distance from reality $f$.

In the process of model development, investigators often make decisions about initial and boundary conditions, and the proper parameterization of the model based on previous knowledge and certain data $y$ that they have at hand. A model would have a set of estimated parameters depending on the available data, which can be expressed as $\hat{\theta}(y)$. It is reasonable to expect that the set $\hat{\theta}$ does not correspond to $\theta_m$, and in fact, in most cases, the configuration of a model would be further apart from the configuration that would minimize its distance to full reality.

In terms of expectations, we are then interested in obtaining an estimator of the distance $I(f, g)$ with respect to the variable of interest $x$ and the available data $y$, expressed as

$$E_y \ E_x[\log(g(x|\hat{\theta}(y)))]. \tag{4}$$

The seminal work of H. Akaike showed that this expectation can be approximated by the log-likelihood function of the fitted parameters given the data

$$\log(\mathcal{L}(\hat{\theta}|y)) - K \approx E_y \ E_x[\log(g(x|\hat{\theta}(y)))], \tag{5}$$

where $K$ is a bias-correction term. Akaike found that under certain conditions, the number of parameters in the model is a good approximation to the bias in using the log-likelihood to approximate the expected relative distance between the model and full reality. Therefore, $K$ is generally assumed as the number of parameters in a model. However, the number of parameters in ESMs is generally very large and unknown for users of model output data, and its inclusion in the estimation of the $I(f, g)$ distance would dominate over the value of the log-likelihood. Therefore, we ignore the bias correction term $K$ and arrive to the expression

$$\log(\mathcal{L}(\hat{\theta}|y)) \approx \hat{E}_{\hat{\theta}}[I(f, \hat{g})], \tag{6}$$

where the $\approx$ symbol is used here instead of an equality because the exclusion of $K$. The key result is that the log-likelihood of a model, configured with parameters and initial conditions consistent with some observed data, is an approximation of its expected distance to full reality.

The non-trivial problem now is to determine a reasonable value of a log-likelihood for a parameterized ESM given some observed data. In contrast to common statistical methods of maximum likelihood estimation (MLE) for simple statistical models, there is no consistent use of MLE methods dealing with the hundreds of parameters in a fully coupled ESM. Also, there is not really any observed data on the gridded format required for comparison with ESM output because measurements are not performed systematically for all points on the surface of the Earth. What we usually have available is data products that are derived from sparse observations and scaled to the terrestrial surface following some data manipulation technique.

Under strong assumptions of linearity, equal variance, and probability distributions from the exponential family (normal, exponential), least-square estimates are identical to maximum likelihood estimates. This is surely not the case for the distribution of output variables from ESMs, but given the absence of any other method for obtaining a log-likelihood function of a parameterized ESM with respect to data, we make here the strong assumption that

$$n \log(\hat{\sigma}^2) \approx \log(\mathcal{L}(\hat{\theta}|y)), \tag{7}$$

where $\hat{\sigma}^2$ is the mean squared deviation between model predictions and some available data product. They are obtained as

$$\hat{\sigma}^2 = \frac{\sum^n \hat{\epsilon}_t^2}{n}, \tag{8}$$

and $\hat{\epsilon}$ are residuals. The lhs in equation (7) is the first component of Akaike's Information Criterion (AIC) for the least-squares case without the correction term $K$. Although not perfect, this approximation to the AIC and the log-likelihood function can be estimated from existing model output from ESMs and some observational data product, provided that both are available for the same spatial and temporal coordinates. Thus, mimicking the definition of AIC, we define for our purposes the distance metric $\mathcal{A}$ as

$$\mathcal{A} := n \log(\hat{\sigma}^2). \tag{9}$$

One characteristic of $\mathcal{A}$ is that it preserves some of the properties of AIC; it can be viewed as a negative entropy (Akaike, 1985), and can be used to compare its value across different models on an absolute scale. In other words, one can compute $\mathcal{A}_i$ for a set of models $i \in [1, \ldots, k]$ and rank them according to their relative difference. One of the models in the set would have the minimum value $\mathcal{A}_m$ and can be considered as the model with the minimum distance to the observations. Furthermore, for

any other model in the set, we can calculate their difference with respect to this 'best' model as

$$\Delta_i := \mathcal{A}_i - \mathcal{A}_m. \tag{10}$$

More importantly, the values of $\Delta_i$ are estimates of

$$E_{\hat{\theta}}[\hat{I}(f, g_i)] - \min E_{\hat{\theta}}[\hat{I}(f, g_i)], \tag{11}$$

i.e., they are estimates of the relative difference between the expected distance between the model and full reality with respect

to the same distance for the model that is closer to full reality (Figure 1). Recall from equation 3 that the expected value for full reality is a constant, therefore its actual value plays no role regarding these values of $\Delta_i$.

Another important contribution of H. Akaike is a method to obtain the likelihood of a model given the data $\mathcal{L}(g_i|data)$ based on the value of $\Delta_i$ for each model. It can be interpreted as the relative strength of evidence for a particular model in the set of models being considered given the available data, and it is expressed as

$$\mathcal{L}(g_i|data) \propto \exp\left(-\frac{\Delta_i}{n}\right). \tag{12}$$

For each model $g_i$ in the set of models, it is also possible to obtain model probabilities, which are weights of the evidence in favor of a model being the model with the lowest $I(f, g)$ distance. These model probabilities or weights can be obtained as

$$w_i = \frac{\exp(-\frac{\Delta_i}{n})}{\sum_{r=1}^{k} \exp(-\frac{\Delta_r}{n})}. \tag{13}$$

Note that $\sum w_i = 1$.

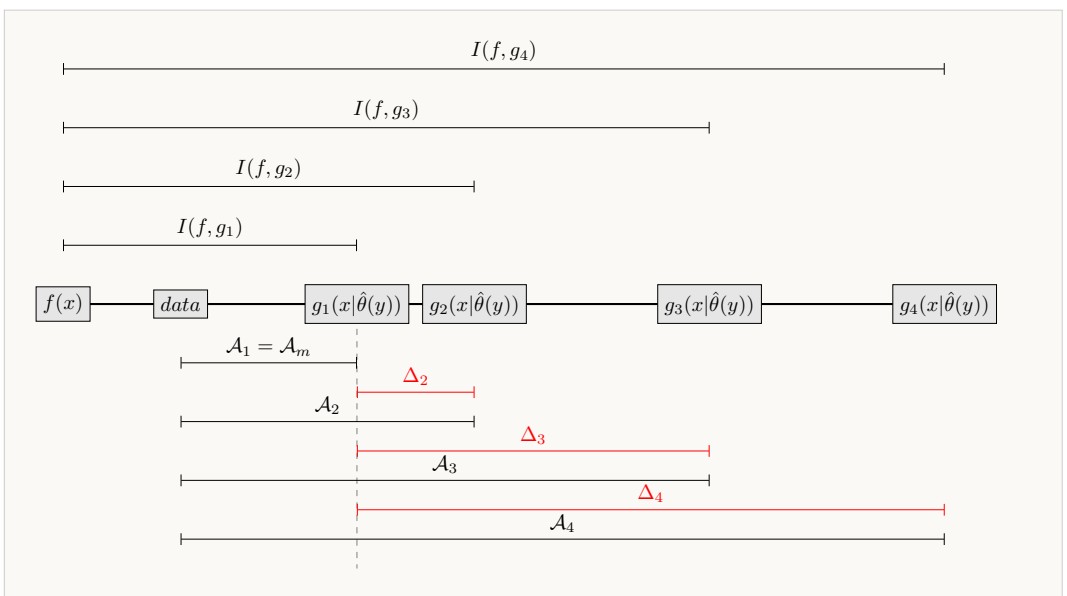

**Figure 1.** Conceptual representation of the approximation of the Kullback-Liebler distance between models and full reality $I(f, g_i)$ by the proposed metric $\mathcal{A}_i$, and relative ranking of models according to the metric $\Delta_i$. Models $g_i$ predict the value of variable $x$ based on a configuration of parameters and initial conditions encoded in the vector $\hat{\theta}$, which is based on some a priori data $y$. In this example, the model with the closest distance to the reference data is $g_1$, so $\mathcal{A}_1 = \mathcal{A}_m$. The relative ranking among models is defined as $\Delta_i = \mathcal{A}_i - \mathcal{A}_m$. Notice that the distance from full reality and data is always constant as well as the distance from full reality to model $g_1$. Therefore, the relative ranking of the models is similar to ranking models based on their Kullback-Liebler distance to full reality.

Combining equation (13) with equations (10) and (9) and after simplification, we arrive at an expression of the weights in terms of the mean squared deviations

$$w_i = \frac{1/\hat{\sigma}_i^2}{\sum_{r=1}^{k} 1/\hat{\sigma}_r^2}. \tag{14}$$

Thus, the weights can be understood as being based on the inverse of the deviations between model output and data product. This expression for the weights (equation 14) is identical to the formula for inverse-variance weighting, which is a maximum likelihood estimator for the mean of a statistical population with independent and Gaussian distribution (Bonamente, 2022). The equivalence between these two expressions for the weights (equations (13) and (14)) emerges only because our choice of distance metric $\mathcal{A}$ to approximate the log-likelihood function, but for other representations of $\mathcal{A}$ these formulas would differ.

With these weights, we can now proceed to compute averages of model predictions according to our strength of belief in the predictions of each model. For a variable of interest $x$, the multi-model average is simply

$$\bar{x} = \sum_{i=1}^{k} w_i \cdot x_i. \tag{15}$$

To obtain the variance of the weighted average, we take advantage of the fact that the variance of an inverse-variance weighted average is equal to the inverse of the deviations (Bonamente, 2022; Kanters, 2022; Rotondi et al., 2022)

$$\text{Var}(\bar{x}) = \frac{1}{\sum_{i=1}^{k} 1/\hat{\sigma}_i^2}. \tag{16}$$

This expression for the variance is consistent with the expression of the weights as in equation (14) where each individual estimate of the inverse-variance for each model is normalized by their sum across all models. Furthermore, the values $1/\hat{\sigma}_i^2$ have a special interpretation from an information theoretic perspective, they are estimates of Fisher's information for distributions of the exponential family (Rotondi et al., 2022). They can be interpreted as the amount of information contributed by model $i$ to the weighted average, so the larger the dispersion of model output from the observations, the lower the contribution of the model to the weighted average.

## 3  Implementation with ESM output and gridded data products

Most numerical output from an ESM is indexed along three coordinate axes, one for time, and two for spatial coordinates. Because the spatial coordinates can be simplified to just one coordinate if we know the mapping from a one-dimensional index to a two-dimensional coordinate space, we define $s$ as the spatial coordinate, and $t$ as the time coordinate. We also define $i$ as the indexing for the models in the set of all considered ESMs. Given these definitions, we consider a model variable $x$ available from one single model for one single grid cell at one point in time as $x_{i,s,t}$. Similarly, we assume that we have a comparable variable derived from observations $y$ from a specific data product $j$ for one grid cell $s$ and one time point $t$ and denote it by $y_{j,s,t}$. The residual for each grid cell and each time point between model $i$ and data product $j$ is given by

$$\epsilon_{i-j,s,t}^2 = (x_{i,s,t} - y_{j,s,t})^2. \tag{17}$$

This equation can be used to produce geographical maps of residuals between one single model $i$ and data-product $j$ for each point in time. An estimate of the mean squared deviation, from the first time point $t_0$ until a final time point $t_f$ would be

$$\sigma_{i-j,s}^2 = \frac{\sum_{t=t_0}^{t_f} \epsilon_{i-j,s,t}^2}{n}, \tag{18}$$

with $n = h(t_f - t_0)$, i.e., the total number of time points, from $t_0$ to $t_f$ multiplied by the time-step $h$. This equation is used to produce one single map (no time points) of mean squared deviations for the grid cells of the model versus the available data product. Now we can compute the distance metric $\mathcal{A}$ as

$$\mathcal{A}_{i-j,s} = n \log(\sigma_{i-j,s}^2). \tag{19}$$

This equation leads to one single map for each model-data product combination. If we repeat this calculation for all models being considered, $i \in [1, \ldots, k]$, we will obtain the set of $k$ number of maps $\mathcal{A}_{i-j,s} := [\mathcal{A}_{1-j,s}, \ldots, \mathcal{A}_{k-j,s}]$. Our purpose now is to identify the grid-cells from this set of maps in which the values of $\mathcal{A}$ are the lowest, and produce one single map as

$$\mathcal{A}_{j,s}^{\min} = \min_{i-j}[\mathcal{A}_{1-j,s}, \ldots, \mathcal{A}_{k-j,s}]. \tag{20}$$

Notice that this map is a combination of all the grid cells that better agree with the data from any of the models. It does not select uniformly all the grid cells of a particular model that perform better, but rather the grid cells from any of the models with minimum $\mathcal{A}$ distance. By doing this, we make sure that we select spatial regions in which particular models perform better than others.

Now we proceed to calculate a set of $k$ maps of differences with respect to this minimum as

$$\Delta_{i-j,s} = \mathcal{A}_{i-j,s} - \mathcal{A}_{j,s}^{\min}. \tag{21}$$

We are now ready to compute a set of $k$ maps of weights as

$$w_{i-j,s} = \frac{\exp(-\frac{\Delta_{i-j,s}}{n})}{\sum_{r=1}^{k} \exp(-\frac{\Delta_{r-j,s}}{n})}, \quad \text{or} \quad w_{i-j,s} = \frac{1/\sigma_{i-j,s}^2}{\sum_{r=1}^{k} 1/\sigma_{r-j,s}^2}. \tag{22}$$

This will result in a set of $k$ maps of weights that will be used to produce a set of $n$ maps (along the time dimension) of the weighted average for the variable of interest as

$$\bar{x}_{j,s,t} = \sum_{i=1}^{k} w_{i-j,s} \cdot x_{i,s,t}, \quad \text{or} \quad \bar{x}_{j,s,t} = \frac{\sum_{i=1}^{k} x_{i,s,t}/\sigma_{i-j,s}^2}{\sum_{i=1}^{k} 1/\sigma_{i-j,s}^2}. \tag{23}$$

The variance of the weighted average is obtained as the sum of the inverse-deviation across models

$$\text{Var}(\bar{x}_{j,s}) = \frac{1}{\sum_{i=1}^{k} 1/\sigma_{i-j,s}^2}. \tag{24}$$

Notice that the weights and the variance of the weighted average are fixed over time (not time-dependent), but they can be used to obtain averages and prediction uncertainties that include the time dimension, even for model output that spans beyond the time interval of the available observations. Thus, the weighted average with interval estimation at one standard deviation can be obtained as (c.f. Rotondi et al., 2022)

$$\bar{x}_{j,s,t} \pm \sqrt{\text{Var}(\bar{x}_{j,s}) + \sum_{i=1}^{k} w_{i-j,s}(x_{i,s,t} - \bar{x}_{j,s,t})^2}, \tag{25}$$

with $t \in [t_0^-, f_f^+]$; i.e., with initial times starting before model output and data product overlap ($t_0^- \leq t_0$) and/or final time after model output and data product overlap ($t_f^+ \geq t_f$). In other words, we can take a smaller period of time when the observational product and the model output overlap to obtain the weights, and then use the weights to average across the entire time span of the available model output.

As a reference, it is useful to obtain the average and the prediction intervals for the equal-weights scheme, where the average is given by

$$\bar{x}_{s,t} = \frac{\sum_{i=1}^{k} x_{i,s,t}}{k}, \tag{26}$$

and the prediction interval can be expressed as

$$\bar{x}_{s,t} \pm \sqrt{\frac{(x_{i,s,t} - \bar{x}_{s,t})^2}{k}}. \tag{27}$$

Notice that the prediction intervals for this equal-weighting case do not include a variance term with respect to an observational product. This is simply because it is irrelevant to include this source of variation when the averaging does not take into consideration the existence of any reference observation. However, for comparing the prediction intervals between the equal-weights and the weighted average, the variance $\text{Var}(\bar{x}_{j,s})$ term in equation (25) adds an undesired source of variability. To make comparisons on equal footing, we thus use a modified version of the prediction intervals for the weighted average as

$$\bar{x}_{j,s,t} \pm \sqrt{\sum_{i=1}^{k} w_{i-j,s}(x_{i,s,t} - \bar{x}_{j,s,t})^2}, \tag{28}$$

which is identical to the expression used in other model weighting approaches previously proposed with climate model ensambles (Sain and Kleiber, 2025, p. 212).

## 4  Ensemble average of land-atmosphere carbon exchange from CMIP6 models

To demonstrate the use of the procedure described above, we computed model weights and a multi-model ensemble average of the net flux between the land and the atmosphere using two different observational products as reference, X-BASE (Nelson, Jacob A and Walther, Sophia et al., 2024) and the Jena CarboScope (Rödenbeck, 2005). For the model ensemble, we used 9 models from the CMIP6 archive that report gross primary production and respiration fluxes as well as net biome production (Table 1).

The X-BASE product is based on the upscaling of eddy-covariance measurements that quantify the net ecosystem exchange (NEE) of carbon dioxide due to the assimilation and respiration of carbon by vegetation and soils. Therefore, this product can be compared with the difference between gross primary production (GPP) and ecosystem respiration (Re) (NEP=GPP-Re) from ESMs. The Jena CarboScope product is based on an atmospheric inversion system that uses mole fraction data of carbon dioxide and predicts net carbon exchange fluxes using an atmospheric transport model. This product is comparable with the variable net biome production (NBP) reported by the ESMs and includes, in addition to GPP and Re, fluxes due to disturbances such as fires and land-use changes.

The minimum distance maps $\mathcal{A}_m$ obtained using the X-BASE and the Jena CarboScope products, showed large differences among each other (Figure 2). However, these maps should not be compared directly because the distance metric $\mathcal{A}$ is an absolute distance metric, and since values of NEP tend to be higher than values of NBP, it is expected that the $\mathcal{A}_m$ distance of the models to the X-BASE product would be higher than the distance of the models to the CarbonScope product (Figure 2). Similarly, when comparing regional differences within any of the two $\mathcal{A}_m$ maps, it is also clear that regions with low carbon fluxes such as the arid and semi-arid regions in Africa, the Arabian Peninsula and Central Australia show the lowest distances to the models. However, these short distances do not necessarily mean that the models perform well in these regions. It just shows that when fluxes are low, the models predict a low distance to the observational product, but this cannot be confused with good performance. For ranking model performance a relative measure such as $\Delta$ (Figures A3 and A4) and the model weights $w$ are preferred.

**Table 1.** Earth system models from the CMIP6 archive used in this study and their relevant features.

| Earth system model | Modelling centre | NEP | NBP | N cycle | Fires | Dynamic vegetation | Land carbon model |
|---|---|---|---|---|---|---|---|
| ACCESS-ESM1-5 | CSIRO | Yes | Yes | Yes (P-cycle) | No | No | CABLE2.4 with CASA-CNP |
| BCC-CSM2-MR | BCC | Yes | No | No | No | No | BCC-AVIM2 |
| CanESM5 | CCCma | Yes | Yes | No | No | Only wetlands | CLASS-CTEM |
| CESM2 | CESM | No | Yes | Yes | Yes | Yes | CLM5 |
| CNRM-ESM2-1 | CNRM | Yes | Yes | Implicit | Yes (Natural) | No | ISBA-CTRIP |
| NOAA-GFDL-ESM4 | NOAA, GFDL | Yes | Yes | No | Yes | Yes | LM4p1 |
| MPI-ESM1-2-LR | MPI | Yes | Yes | Yes | Yes | Yes | JSBACH3.2 |
| NorESM2-LM | NCC | Yes | Yes | Yes | Yes | No | CLM5 |
| UKESM1-0-LL | UK | Yes | Yes | Yes | No | Yes | JULES-ES1.0 |

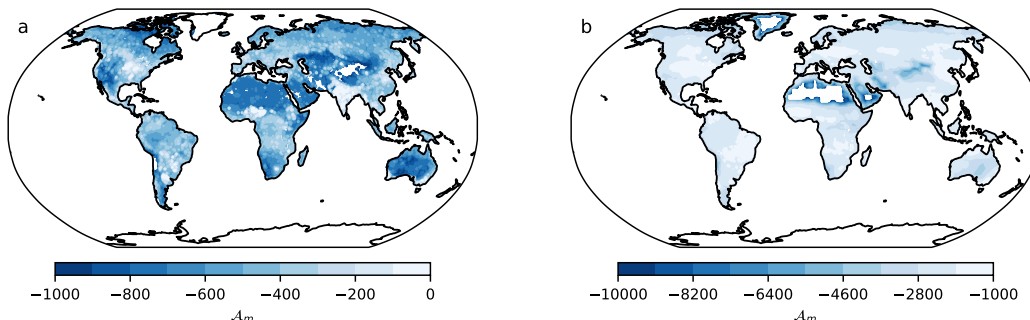

**Figure 2.** Minimum distances ($\mathcal{A}_m$) between (a) NEP from CMIP6 ESMs and X-BASE, and (b) NBP from CMIP6 ESMs and the Jena CarboScope product. More negative values in darker colors indicate smaller distances (values in logarithmic scale), representing larger similarity between the ESMs and the observational products. Note that the numerical scales between a and b are different.

The weights obtained for each model provide a relative ranking of the models with respect to their distance to the observational product, and serve as a suitable metric to assess model performance. The values obtained differed considerably between the NEP and the NBP reference (Figures 3 and 4), which shows that model weights may change substantially depending on the observational product being used as a reference. Furthermore, for each model, it is clear that there are regions that perform better or worse than others in comparison to the observational product. For example, the MPI-ESM1-2-LR model performs consistently poorly in the Amazon region, in tropical Africa and North America, but performs relatively well in Europe and northern Eurasia. Other models also show consistent spatial patterns of good or poor performance, indicating that the weights do not capture randomly spaced grid-cells, but aggregated regions where the models tend to perform consistently in either direction with respect to the observations. These maps of weights also show that there is no one single model that performs best everywhere, or contrastingly, one single model that performs worse everywhere.

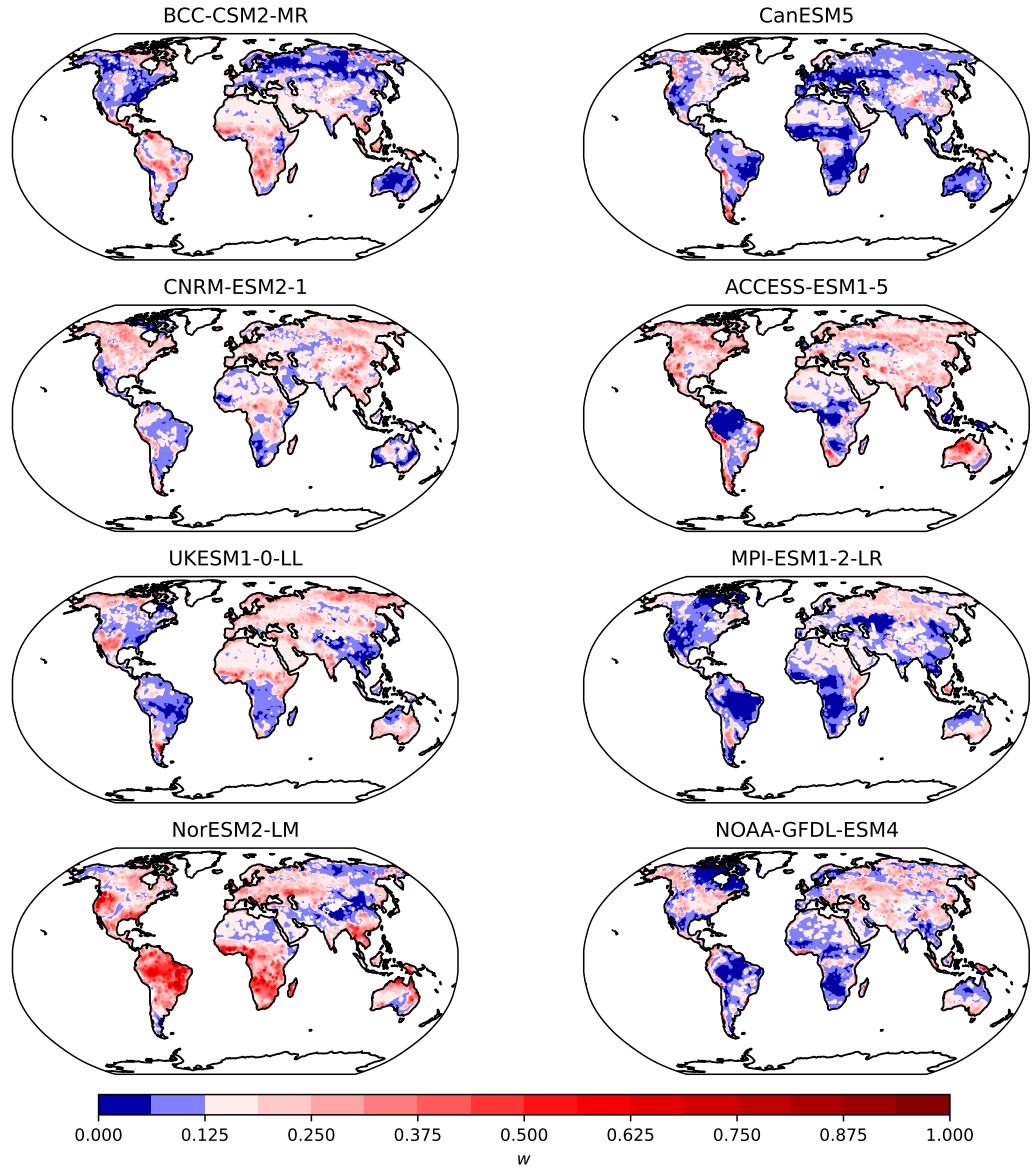

**Figure 3.** Weights $w$ of CMIP6 models for the variable NEP with respect to the X-BASE observational product. The diverging color palette is centered at a value of 1/8, indicating whether a model contributes more or less than the equal weight of 1/8 from the $k = 8$ models. For each grid cell, the sum of the weights of the 8 models adds up to 1.

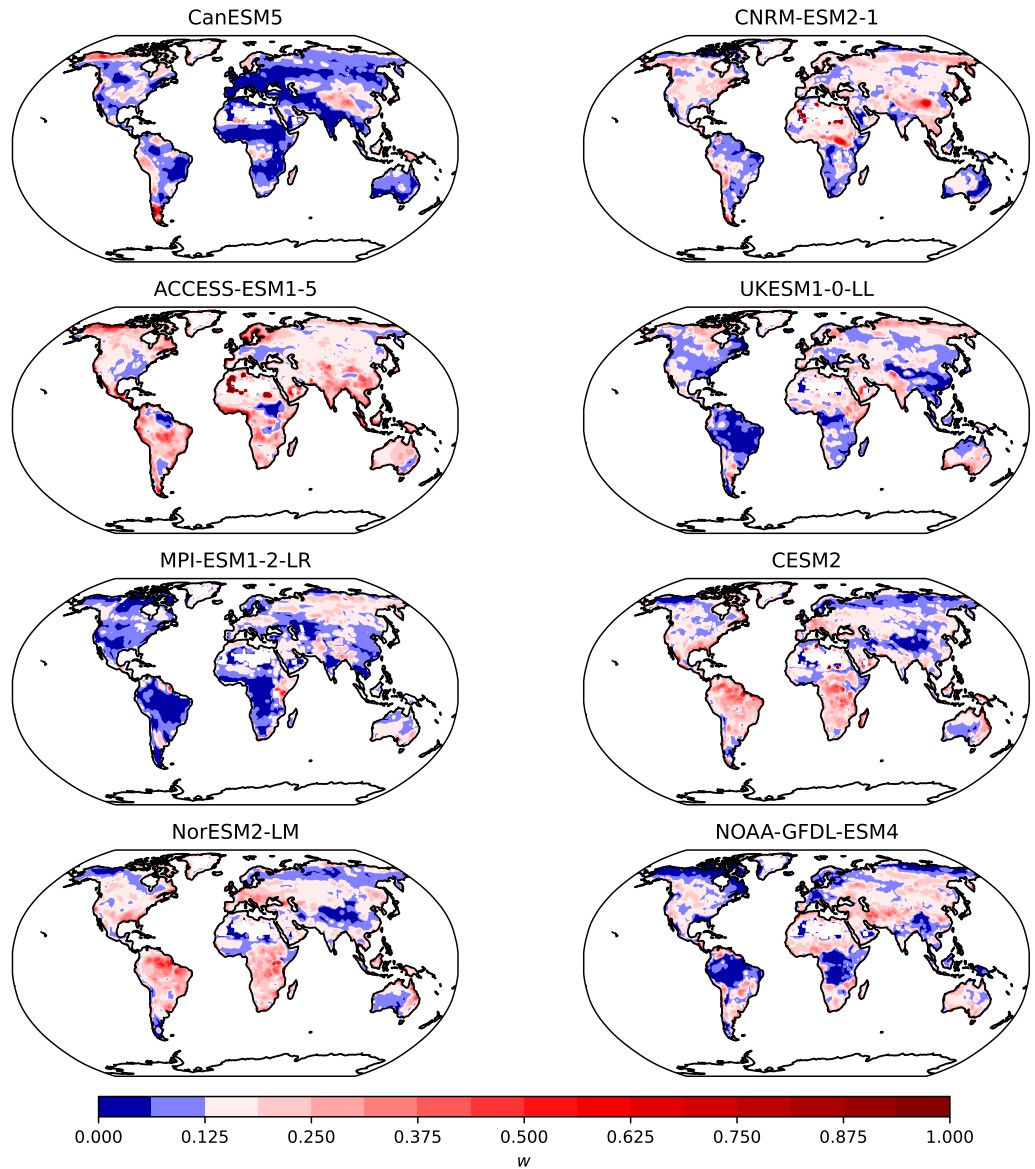

**Figure 4.** Weights $w$ of CMIP6 models for the variable NBP with respect to the Jena CarboScope observational product. The diverging color palette is centered at a value of 1/8, indicating whether a model contributes more or less than the equal weight of 1/8 from the $k = 8$ models. For each grid cell, the sum of the weights of the 8 models adds up to 1.

The values of weights for each grid cell were combined with the predictions of the variable of interest for each model to produce weighted averages at the grid cell level for each time step, and then summed across grid cells to obtain time series (Figure 5). The obtained results show that the weighted average of NEP is consistently lower than the arithmetic average (38.2%), mostly because the influence of models that make predictions with values much higher than the observational product have much less weight in the weighted average (Figure 5a). In fact, the models that predict the highest values of NEP, namely NOAA-GFDL-ESM4 and ACESS-ESM1-5, make predictions well outside the uncertainty range obtained for the weighted average, indicating the small contribution that these models make to the obtained average. The prediction uncertainty was also considerably smaller for the weighted average, approximately 38.2% lower than the uncertainty for the arithmetic average.

For the variable NBP, the arithmetic and the weighted average are relatively close to each other for the entire simulation period (Figure 5b). In this case, most models make predictions close to each other and, therefore, they contribute more evenly to the weighted average. Nevertheless, the obtained uncertainty range is lower for the weighted average (39.1%), indicating that those models closer to the observational product have more weight in terms of both the average and its variance, and therefore help to reduce overall uncertainty in the predictions.

For both variables, NEP and NBP, the estimate of uncertainty with our proposed approach is also significantly lower than the uncertainty based on equal weights (Figure A2). We obtained a much lower level of uncertainty for the weighted average because the method applies probabilities or strengths of belief to the different models and grid cells, and therefore this averaging procedure increases confidence in the inferred multi-model average.

However, the prediction intervals in Figure 5 do not include the inter-model variance term $\mathrm{Var}(\bar{x}_{j,s})$ from equation (25); i.e., only the deviation of the models with respect to the averages excluding the deviation of the models from the observational product. When combining both sources of uncertainty, the overall variance is much larger for the weighted average (Figure A2). For the arithmetic average, this source of variation is generally not included because it is irrelevant in this context to compare the models with observations, but in case this source of uncertainty would be included, the prediction intervals of the arithmetic average would be much larger than for the weighted average.

It is also important to note that even though the observational products are only available for a short period of time, we used the obtained weights for the entire period of the simulations of the ESMs under the assumption that a model that performs well during the period in which observations are available, should be able to perform well for other periods. This assumption is obviously questionable, and probably inadequate for periods of time much beyond the time range of available data. However, we still believe that this assumption is better than to assume that all models are equally reliable for all periods of time as it is implicitly assumed with an equal weight approach.

## 5  Discussion

Although other authors have proposed methods to obtain weights and multi-model averages from the predictions of ESMs (Tebaldi and Knutti, 2007; Merrifield et al., 2020, and references therein), we presented here an approach based on information-theoretic concepts that is easy to implement and can help to improve inferences from ESMs minimizing biases and reducing

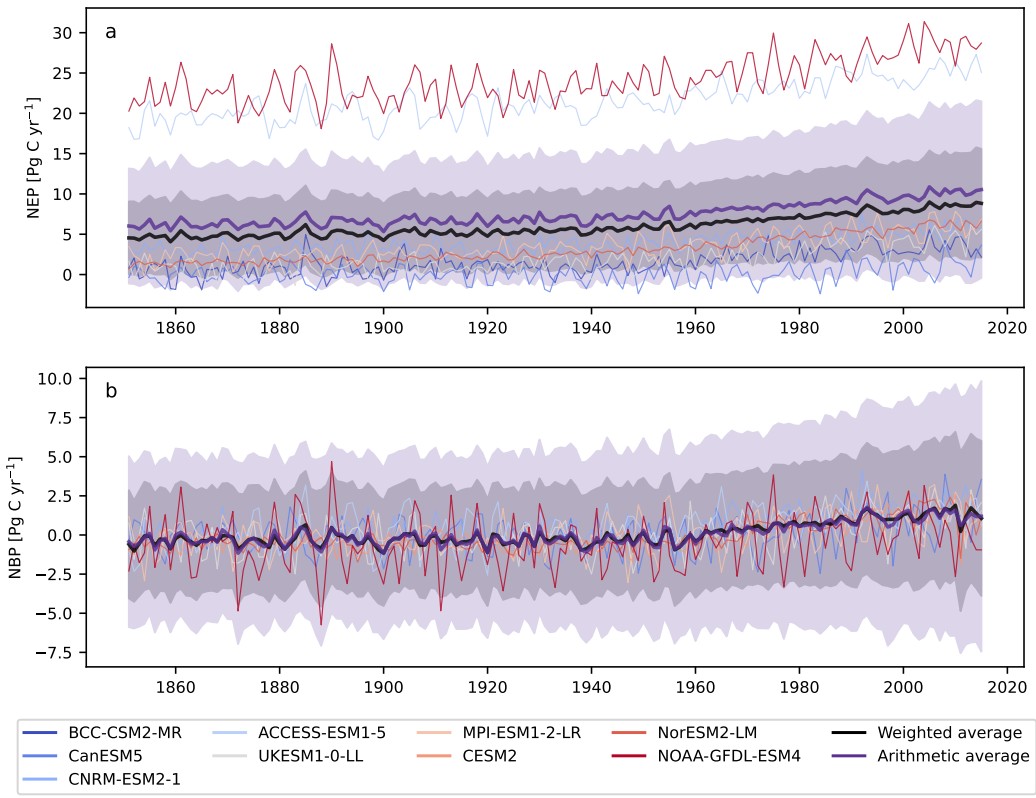

**Figure 5.** Annual time series of (a) NEP and (b) NBP from individual CMIP6 ESMs obtained as arithmetic averages (purple lines) and the weighted averages using the method proposed here (black lines). Uncertainty ranges are expressed as the average value $\pm$ the value of $\sigma$ including only the second term from equation 25. For the arithmetic average, the weights for the computation of uncertainty are $w_i = 1/8$.

uncertainties. Most previously proposed methods have focused on the problem of probabilistic climate change projection taking into account the variability around the ensamble average (Giorgi and Mearns, 2002, 2003; Tebaldi et al., 2005; Knutti et al., 2017). Therefore, many of these methods lack a level of generality adequate for other problems of interest. In addition, many of the previously proposed methods include a step in which a generalized linear model is fit to the model output to obtain weights for specific regions or periods of time (Greene et al., 2006). For approaches that use a Bayesian approach (e.g. Tebaldi et al., 2005), it is necessary to specify a family of prior distributions for the mean and variance, which adds a layer of complexity and uncertainty for obtaining the model weights.

As opposed to these previous methods, our approach makes no assumptions of a generalized model and/or prior distributions. Additional advantages of this method over others include: (1) a theoretical foundation based on concepts from probability and information theory (Akaike, 1974, 1981; Anderson, 2007; Burnham and Anderson, 2002). (2) The weights have a relevant interpretation, they are evidence in favor of a model prediction having the smallest distance to full reality, even though comparisons are only performed with respect to an uncertain observational product. For the particular choice of $\mathcal{A}$ as the logarithm

of the mean squared deviations, the weights have a straight forward interpretation: the models with less deviation from the observations contribute more to the weighted average. (3) The calculation of weights is based on spatial performance of model output with respect to observations, and not a uniform weight for all grid-cells of a model.

For the particular choice of the distance metric $\mathcal{A}$ as the logarithm of the mean squared deviation between model output and observational product, we obtained weights that are similar to an inverse-variance weighting scheme. This result is expected because the inverse-variance weighted average is an efficient maximum likelihood estimator of the mean of a statistical population, which corresponds to our choice of the mean squared deviation as a log-likelihood function. But we know that this choice of log-likelihood function is not ideal, and should be replaced if other approaches for MLE are available, which would result in an expression for the weights different than the inverse-variance weights. Nevertheless, the use of inverse-variance weighting is an intuitive and easy approach to apply to ESM ensembles, and it is also common in other fields such as in meta-analyses (Hartung et al., 2008; Kanters, 2022) and in biomedical studies (Mansournia and Altman, 2016).

However, as with other approaches, some weaknesses should be acknowledged and should be improved in future research. These are: first, lack of a log-likelihood function for the assessment of model-observation distances. Although such a function is difficult to obtain given the process of development and parameterization of ESMs, it is still desirable to obtain an unbiased estimator of the log-likelihood of a parameterized model with respect to available data. We are not aware of another method that could be used to replace the simple log of square residuals used here, but it is also important to point out that other measures of distance used in other methods apply mostly the squared residuals as a distance metric. Therefore, our method offers a small theoretical improvement over previous approaches based on the theoretical knowledge that, under certain assumptions, the logarithm of square residuals is a first-order approximation to a maximum likelihood estimator.

Second, other authors have raised concerns over the issue of model independence (Knutti, 2010; Knutti et al., 2010), a problem that we do not address explicitly here. They argue that many models share the same base code or are based on the same underlying principles, and cannot be treated as completely independent estimates for obtaining an unbiased average. In particular, the method of Knutti et al. (2017) produces weights that penalize a model according to its prediction distance to that of other models. We think this is a valid concern, particularly when weights are obtained for the aggregated (sums or averages) across all grid cells of a model. However, the implementation of similar processes in two models but with differences in other components such as its climate sensitivity may lead to very different predictions. For instance, CESM2 and NorESM2-LM share the same land vegetation model, CLM5 (Table 1). Although the spatial distribution of the weights for NBP with these models tends to correlate well, the correlation is not uniform across all grid cells and is mostly below 90% (Figure 6). This implies that a shared component of a model can interact with other non-shared components, resulting in different predictions that should be treated differently for the calculation of weights.

It is also important to note that, in the ideal case in which we would have a perfect understanding of Earth system processes, all mathematical models representing these processes would converge to the same predictions. Therefore, it is still debatable whether models that agree with each other because they have a common representation of underlying processes should be penalized. For this reason, we refrain from introducing a penalization term to our computation of weights, but we acknowledge

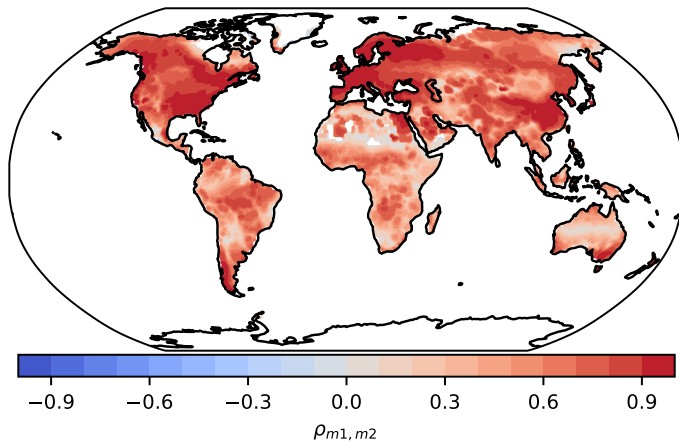

**Figure 6.** Pearson correlation coefficients $\rho$ of variable NBP between CESM2 ($m1$) and NorESM-LM ($m2$). These two models share the same land model (CLM5), but due to differences in other components of the model, the predictions of NBP are not similar and often correlate with values of $\rho < 0.9$.

that this is an issue that deserves more theoretical work. In particular, from an information-theoretic point of view, the problem should be a dressed in terms of the mutual information shared among models (Majhi et al., 2023).

A third issue that deserves more attention is the lack of penalization for model complexity in the approach we propose. The original work proposed by H. Akaike is very well known for the introduction of a penalization term due to the number of parameters in the model, a penalization well supported by mathematical theory and philosophy of science (Akaike, 1974, 1981). However, the scientific trend in ESM development is the addition of increased levels of detail supported by increased computational power (Held, 2005, 2014). We do not have information on the total number of parameters used in each ESM, but we believe it should be on the order of $10^2 - 10^3$. Therefore, adding a penalization term as in the traditional form in the computation of AIC would lead to differences among models that are dominated by differences in their number of parameters, obscuring differences in model distances with respect to observations. But ignoring the penalization due to model complexity, as we do in our approach, implies that we continue ignoring the tension between model complexity and understanding, and focus exclusively on model performance. We believe this a topic that deserves much more theoretical attention and should be addressed in future improvements on the approach we propose.

## 6  Conclusions

We proposed here an approach to obtain probabilities of model performance with respect to available observational products, and to derive weights of evidence in favor of a model being the best from a set of available models. These weights are not

constants for a particular model, but are obtained at the grid-cell level for each model. They provide estimates of the relative likelihood that a model performs well at a particular grid cell and therefore can be used as the weight of evidence for a model performing well in a particular location. We believe this probabilistic interpretation, grounded on solid concepts from information theory, provides advantages over other methods and can be of real practical use for making inferences of average behavior in Earth system science.

Using this approach for obtaining ensemble averages of the variables NEP and NBP from models participating in CMIP6, we found that our proposed weights can significantly reduce bias when a small number of models make predictions further away from a reference observational product and all other models in the ensemble. The prediction uncertainty for the weighted average is also smaller than the uncertainty of the arithmetic average. Overall, the approach helps to increase confidence in inferring spatial and temporal behaviors from multiple models.

*Code and data availability.* The exact version of the code used to produce the results used in this article is archived on repository under DOI: 10.5281/zenodo.15167573 (Sierra and Muñoz, 2025), as well as the model weights necessary to compute model averages (Sierra and Muñoz, 2025).

## Appendix A: Supplementary figures

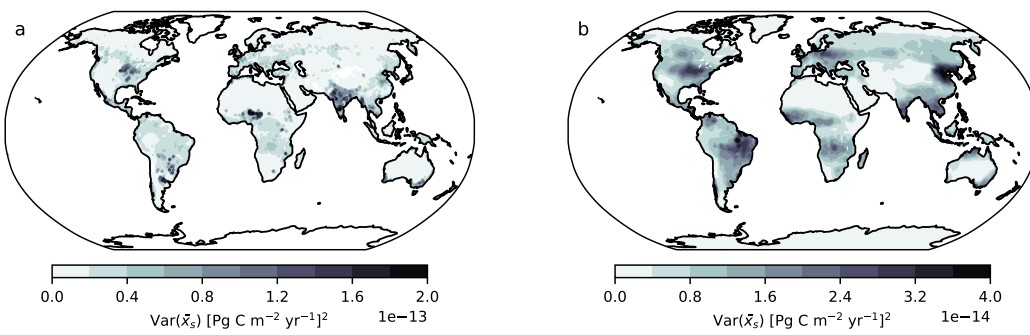

**Figure A1.** Variance of the weighted average from the CMIP6 ESMs outputs of (a) NEP using X-BASE and (b) NBP using Jena CarboScope. The variance is calculated from equation (24).

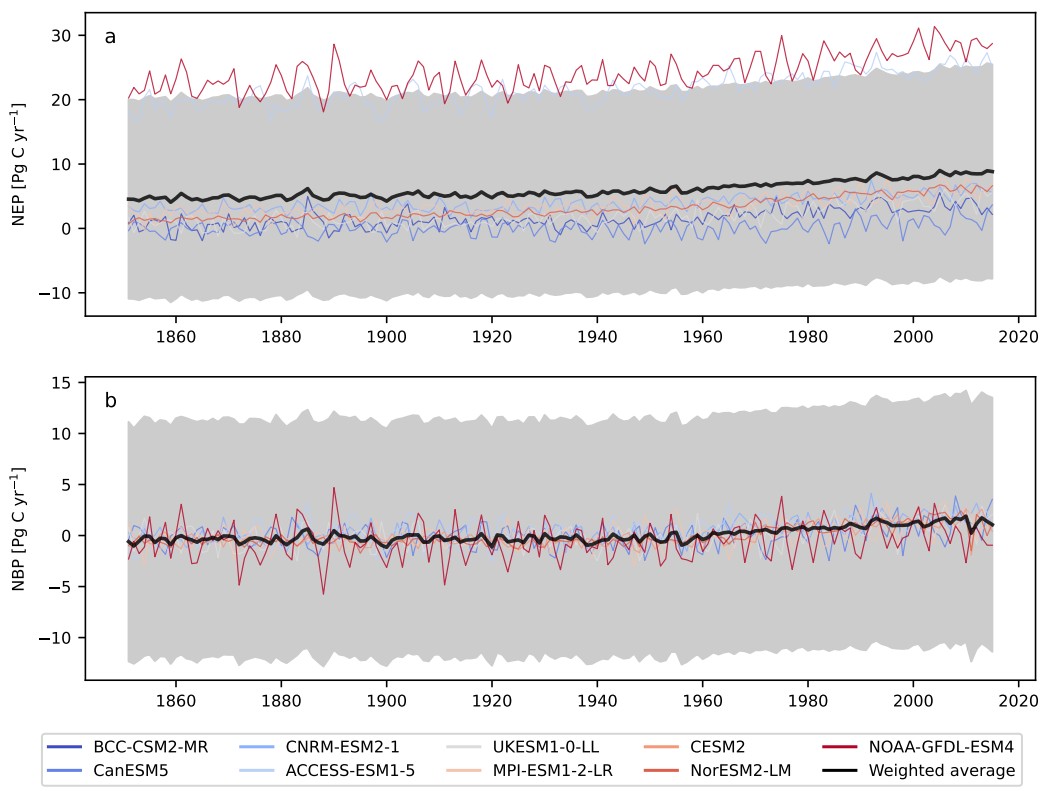

**Figure A2.** Annual time series of (a) NEP from CMIP6 ESMs and X-BASE, and (b) NBP from CMIP6 ESMs and the Jena CarboScope product obtained as the weighted averages using the method proposed here (black lines). Uncertainty ranges are expressed as the average value $\pm$ the full value of prediction uncertainty from equation (25).

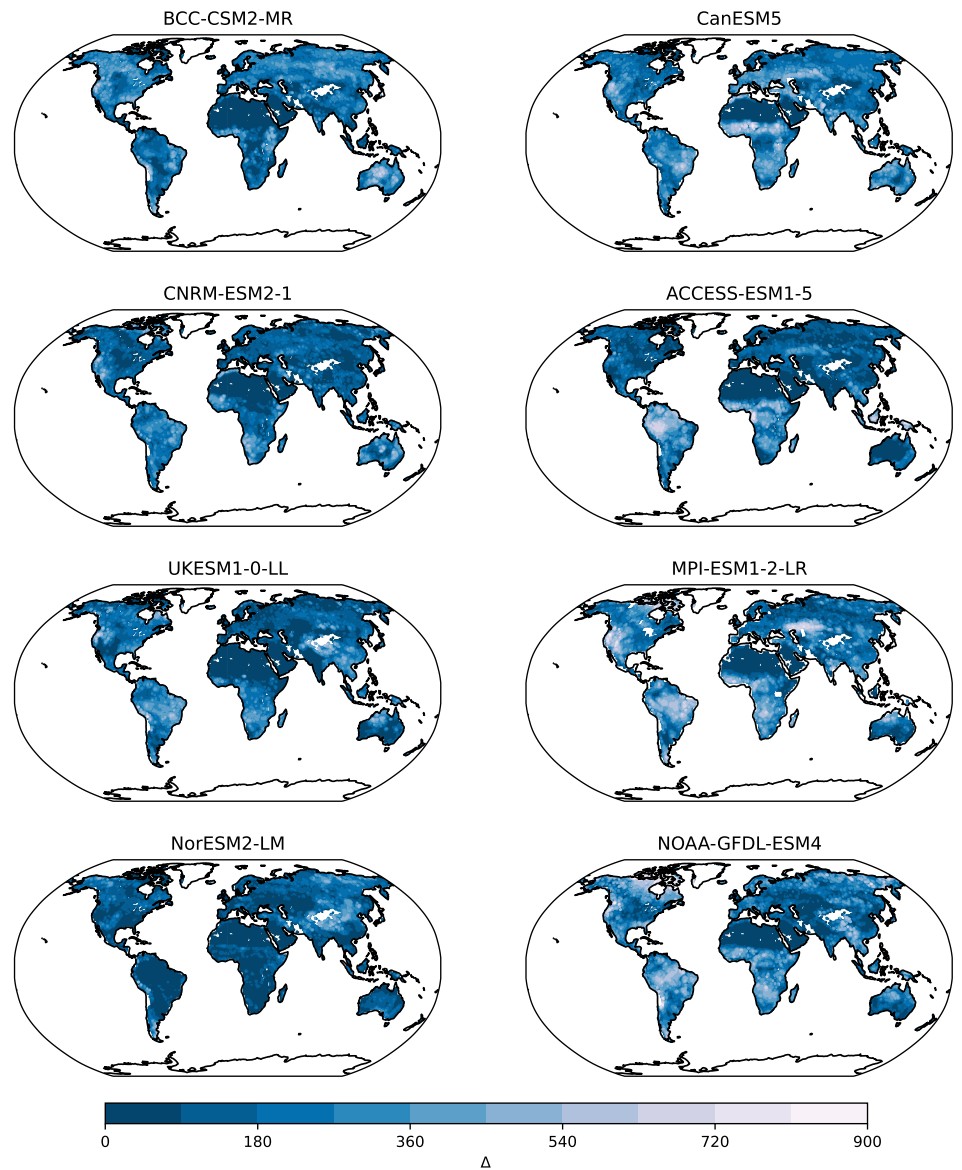

**Figure A3.** Maps of differences $\Delta$ with respect to minimum distances ($\mathcal{A}_m$) of CMIP6 models for the variable NEP and the X-BASE observational product. Small numbers in darker colors indicate smaller distances, representing a larger similarity between ESMs and X-BASE.

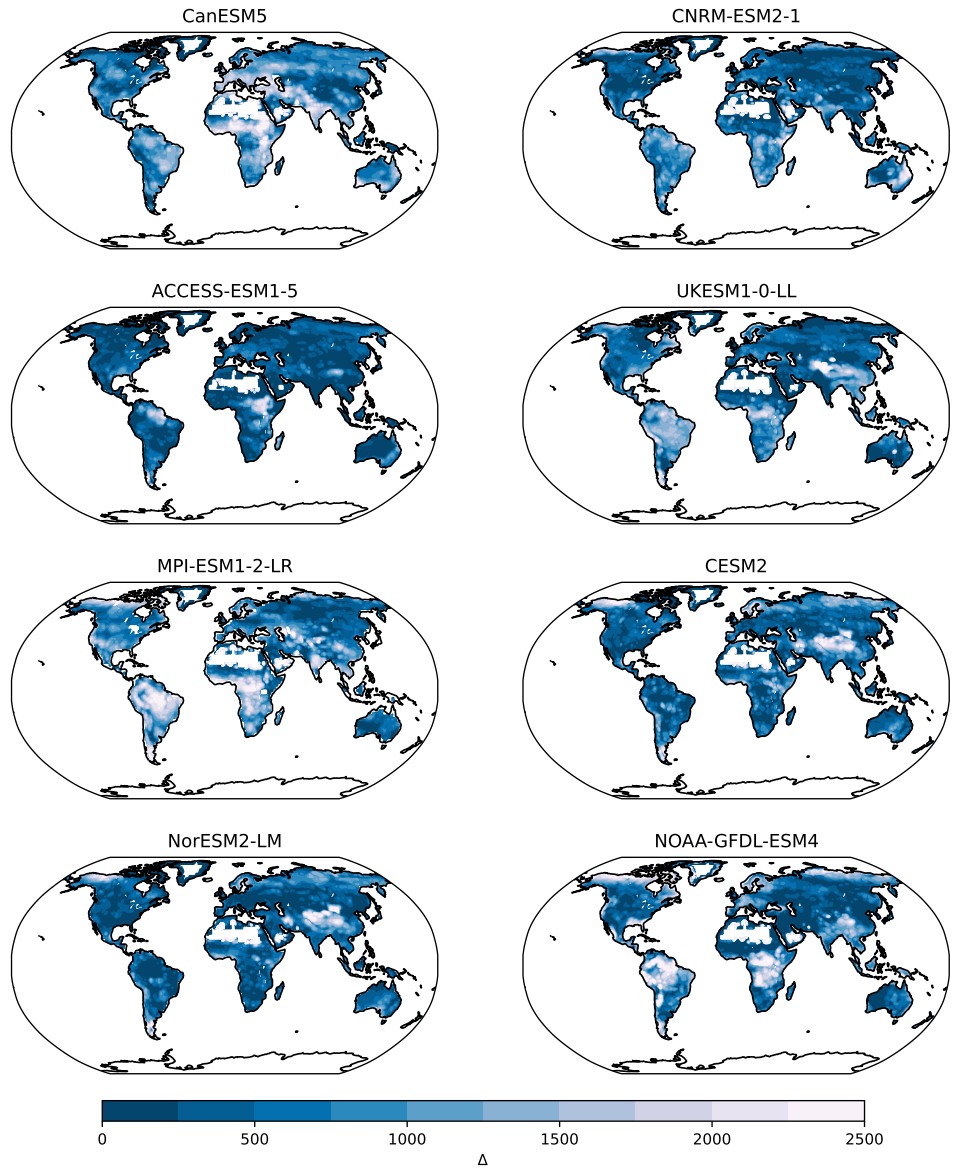

**Figure A4.** Maps of differences Δ with respect to minimum distances ($\mathcal{A}_m$) of CMIP6 models for the variable NBP and the Jena Carbo-Scope observational product. Small numbers in darker colors indicate smaller distances, representing a larger similarity between ESMs and CarboScope.

*Author contributions.* CAS: Conceptualization, methodology, writing–original draft. EM: Data curation, investigation, software, visualization, writing–review & editing.

*Acknowledgements.* CAS acknowledges core funding from the Max Planck Society. EM received funding from the European Union's Horizon 2020 research and innovation programme under the Marie Skłodowska Curie grant agreement No 101110350.

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
