# Peer review of "An information-theoretic approach to obtain ensemble averages from Earth system models"

_EGUsphere, 2025_

## Referee Comment (RC2)

**Review of Manuscript**

**An information-theoretic approach to obtain ensemble averages from Earth system models'**
By C. Sierra and E. Munoz

Dear Editor,

I have reviewed the manuscript. My conclusions and comments are as follows:

**1. Scope**

The article is within the scope of GMD.

**2. Summary**

In their manuscript, the authors propose a method for multi-model inference, in particular for the task of estimating quantities of interest as single values from ensembles of earth systems models (ESMs), with the particular challenges of the models i) being parameterized in non-MLE ways, ii) having a large but unknown number of parameters and iii) high dimensionality. The approach seeks to go beyond simple ensemble averaging by including a measure of model performance in the weighted combination. The method is based on information concepts in general and work by Akaike, Burnham and Anderson in particular, based on the key insight that the log-likelihood of a model trained on a set of observation data is equivalent to its expected Kullback-Leibler divergence, which is a universal measure of distance between a reference pdf (here: the observations) and a model pdf thereof. The log-likelihood therefore can be used to derive suitable weights when taking the average of the outputs of an ensemble of ESMs. The challenge is the estimation of the likelihoods. Under certain conditions they can be estimated by the variance of the residuals (= difference of model and reference truth). For ESM output these conditions are not met as the authors themselves state, but they use this approach for the lack of alternatives. The method then consists of calculating, grid-by-grid (to acknowledge spatial patterns of model performance) for each model the variance of the residuals against some observational product, selecting the best among the models and calculating the performance difference (delta) of the remaining models against this reference. Following Akaike, the exponential of delta is then used to obtain weights for each model, which then allows calculating a single-valued model ensemble average and related variances as a measure of estimation uncertainty. The authors then demonstrate their method at the example of 9 ESMs and two related observational products. A comparison to simple equal-weight ensemble averaging shows that i) they may differ substantially in the presence of poor-performing models and ii) that uncertainty estimates are generally much lower for the weighted average approach.

**3. Evaluation**

Overall, the manuscript provides a solution for an important task – averaging of ESM ensembles - that is both well-grounded in theory and easy to apply. The (sometimes strong) assumptions are clearly stated, and the method overall is well presented in terms of derivation and examples.

A few questions remain:
1) Calculation of models weights. In eqs (7) – (15), I take it that index "i" goes over all models except the best performing model ("m"), which serves as the reference. Does that mean the best performing model will not be included in the weighted average? Please clarify.
2) Patterns of model performance in space and time (see also line 180): The authors resolve spatial patterns of model performance by calculating model weights grid-by-grid over all points in time. This inherently assumes temporal invariance of relative model performance, which clearly is not the case. Therefore I wonder if it would not be more appropriate to

derive model residuals (and deltas) from space-time regions rather than time-regions alone. Please comment.

3) In Sect. 4, simple averaging is used as a benchmark to compare the weighted averaging proposed by the authors. While e.g. Fig. 5 clearly show differences among the methods, it is not clear if the weighted average really provides the better (in terms of smaller disagreement from observations) estimate that the simple average. I am quite sure this will be the case, but please add and discuss the related numbers.

My overall recommendation is to publish after these minor points have been suitably addressed.

Yours sincerely,

Uwe Ehret

---

## Author Response (AR1)

**Max–Planck–Institut für Biogeochemie**
Max Planck Institute for Biogeochemistry

[Figure]

MPI für Biogeochemie · Postfach 10 01 64 · 07745 Jena, Germany

Olivier Marti
Editor
Geoscientific Model Development

**Dr. Carlos A. Sierra**
Tel.: +49-(0)3641-57-6133
csierra@bgc-jena.mpg.de

10th August 2025

Dear Editor,

Thanks for your editorial support with this article. We have provided answers to all comments from the reviewers and prepared a new version of the article clarifying terminology, expanding a set of equations, and making connections to other approaches. In particular, we provide now equations that are equivalent to inverse-variance weighting methods as suggested by reviewer 1.

You can find point-by-point answers to all reviewers' comments below. These are the same answers that were already provided in the discussion forum.

**Reviewer 1**
*This is a nice study that proposes an information-theoretic rationale for weighting ESM outputs when computing multi-model average projections. The approach constructs weights from the divergence between each ESM's output distribution and the observed-climate distribution, thereby rewarding models that align more closely with an observational product. The method is demonstrated on an ensemble of eight CMIP6 models to project net ecosystem exchange of $CO_2$ and net biome production, with weighting schemes calibrated against observational datasets. I found the study well written, with a clear and intuitive presentation of the information-theoretic background. These concepts are often missing from discussions of climate-model post-processing, and it is refreshing to see them used here. I also enjoyed learning about the connection between cross-entropy and AIC. I have a small quibble with calling the KL divergence a distance, but I will not press the point because the term likely helps build intuition.*

We thank the reviewer for the positive and constructive evaluation of our article. We agree with the reviewer's concern about our use of the term 'distance', and acknowledge that 'divergence' is a more appropriate term. Most introductory textbooks on information theory make this clarification, but we used the word in our original version assuming no previous knowledge of readers on the details of distance metrics (norms) in mathematics. However, we see now the potential source of confusion given that a portion of the readers of this article might be

Max-Planck-Institut für
Biogeochemie
Hans-Knöll-Straße 10
07745 Jena
Germany

Tel.: +49-(0) 3641 / 57−6110
Fax.: +49-(0) 3641 / 57−6110
http:// www.bgc-jena.mpg.de

Direktorium
Markus Reichstein (Managing Dir.)
Susan Trumbore
Sönke Zaehle
ID-Nr. DE 129517720

familiar with the mathematical definition of metric and norm.

To address this issue, we added in section 2 a paragraph clarifying the difference between 'distance' and 'divergence', and point out that for the purpose of this article, with treat both terms as synonyms.

*Although I am not deeply familiar with all work on combining ESM outputs, my understanding is that another common strategy is to reward models that (i) simulate today's climate well and (ii) remain close to the ensemble consensus for future change. The manuscript cites earlier work (e.g. Tebaldi & Knutti) at several points, but a fuller discussion of how existing methods compare would be valuable. Readers will want guidance on when this weighting scheme should be preferred and why.*

The literature on multi-model ensemble averages is relatively rich, and there are more approaches than the one mentioned by the reviewer here. It is not our intention to provide here a literature review on this topic, as other reviews already exist. Nevertheless, we added a paragraph in the Discussion section in which we briefly mention the type of available approaches as reviewed by Tebaldi & Knutti (2007), with additions of more recent and relevant references.

*In addition, I have a minor comments/questions that I hope the authors will be able to address before this is considered for publication. L95-100 : could the authors expand on why the approximation dismissing K is appropriate? I know this is discussed later in the manuscript as a limitation of the proposed method, but I think it would be useful to also have an argument at this point on why that's a reasonable approximation to start with.*

We modified this section by first providing a version of equation 5 that includes $K$, followed by the approximation version without $K$. We explain the reason for not including $K$ here following the same arguments provided in section 5.

*L105-110 : "but given the absence of any other method for obtaining a log-likelihood function of a parameterized ESM with respect to data" I would recommend nuancing this statement. There exists methods out there that allow to model loglikelihood functions (e.g. variational approaches). This doesn't diminish the proposed approach, since it might be the simplest first step to take, and in the Occam's razor philosophy, it makes sense being explored and worthy of a publication.*

We modified this paragraph based on the reviewer's suggestion. It is true that models that use some parameterization schemes such as the 4D-var method and its variants, provide the possibility to obtain a likelihood function. However, these approaches are often used in one component of the model, and not necessarily to parameterize a fully-coupled ESM. Nevertheless, it may be possible to use results from these optimization approaches to add some information on the likeliihood function for some component of the ESM.

*Eq 13 : Am I correct in saying that the weights end up being $w_i = 1/\sigma_i / \sum 1/\sigma_i$? I think it would be useful to explicitly include this in the manuscript. The current presentation aims for a greater level of generality in its formalism, which is commendable, and could apply to any choice of distance metric A. However, for the particular choice made by the authors here, the expression of wi simplifies a lot and becomes very interpretable : we simply give more weight to model that have better least square agreement with the observational product.*

We thank the reviewer for this important comment. Indeed, for our particular choice of the metric $A$, the weights can also be expressed as the inverse of the variance between model output and observations. In fact, in the literature on maximum likelihood estimation, inverse-variance weighting emerges as an efficient estimate of the mean for populations in which the variances are known and the mean is unknown. This is an interesting connection between the information-theoretic approach and the maximum-likelihood theory, which converges to the formulas of inverse-variance weighting for our choice of metric $A$. Based on this result, we modified some of the presentation of the theoretical results, making a better link to the maximum likelihood theory, showing alternative formulas for the weights based on the inverse-variance equations, and presenting a simpler formula for the overall variance. In addition, we found that inverse-variance weighting methods are common in meta-analyses and in the biomedical literature, so we added a few sentences in the Discussion showing these alternative use of the method.

*Eq 15 : Is this supposed to be a definition of the uncertainty or the variance of $\bar{x}$? If the latter, I don't understand how it is derived, if the former I would suggest not using $\bar{x}$ as a subscript.*

We modified and clarified the representation of variance and uncertainty. Given the previous result on the weights being identical to inverse-variance weighting, we used this result to provide a formal definition of variance. To represent uncertainties, we changed the equations to express them as predictions intervals given that this is a more appropriate way to express the uncertainties for predictions outside the time range where model output and observations overlap.

*With these points addressed, I believe the paper will make a valuable contribution and be ready for publication.*

*Just out of curiosity : I appreciate that the distance $A$ is only interpretable as a relative metric. I've nonetheless always been curious about its interpretation in "informational units". What I mean is that Shanon entropy measure information in bits, which can be argued to be an intepretable unit. I guess this doesn't translate immediately here since in the continuous setting we're using the differential entropy which is homogeneous to x. But have you thought of ways to make its values as an absolute metric interpretable?*

The original article of Kullback & Liebler (1951) helps to arrive at an interpretation of our proposed metric $A$, and the model differences $\Delta$. It is important to keep in mind that the origin of the KL divergence was in the context of the statistical problem of discriminating between two populations. Therefore, the interpretation of the KL distance in this context is of the information available to distinguish between two statistical populations (Kullback & Libler 1951, pg. 80). More generally, one can also interpret KL divergence as the information available to distinguish between two probability distributions. This information would be measured in bits or nats depending on the base of the logarithm. Regarding our definition of $A$ and its interpretation. $A$ is based on the log-likelihood between model predictions and observations to approximate the KL divergence, so we can interpret $A$ as the available information to discriminate between the distribution of the model and the observational product. We used the base $e$ logarithm, so $A$ is measured in nats. Now, the values of $\Delta_i$ are differences among the values $A_i$ for

individual models and the values of $A$ of the model with the lowest divergence to observations. Therefore, we can interpret the values $\Delta_i$ as the information available to discriminate between each individual model and the value of the model with the lowest divergence with the observations, measured in units of nats.

**References**

Kullback, S. and Leibler, R. A.: On Information and Sufficiency, The Annals of Mathematical Statistics, 22, 79–86, http://www.jstor.org/stable/2236703, 1951.

Tebaldi, C. and Knutti, R.: The use of the multi-model ensemble in probabilistic climate projections, Philosophical Transactions of the Royal Society A: Mathematical, Physical and Engineering Sciences, 365, 2053–2075, https://doi.org/10.1098/rsta.2007.2076, 2007.

**Reviewer 2**

*Overall, the manuscript provides a solution for an important task – averaging of ESM ensembles - that is both well-grounded in theory and easy to apply. The (sometimes strong) assumptions are clearly stated, and the method overall is well presented in terms of derivation and examples.*

*A few questions remain: 1) Calculation of models weights. In eqs (7) – (15), I take it that index "i" goes over all models except the best performing model ("m"), which serves as the reference. Does that mean the best performing model will not be included in the weighted average? Please clarify.*

The index $i$ runs over all models, including the best performing model $m$. For the calculation of the weights, the best performing model has a value $\Delta_m = 0$, which implies that the value of the numerator in the computation of the weights (eq. 13) is equal to 1, and for all other models the value in the numerator is less than 1.

*2) Patterns of model performance in space and time (see also line 180): The authors resolve spatial patterns of model performance by calculating model weights grid-by-grid over all points in time. This inherently assumes temporal invariance of relative model performance, which clearly is not the case. Therefore I wonder if it would not be more appropriate to derive model residuals (and deltas) from space-time regions rather than time-regions alone. Please comment.*

In principle we agree in that model performance should account for spatial and temporal covariations. However, it is not trivial to include these covariations in our information- theoretic approach because they have to be treated in the context of mutual information. The same reasoning applies for covariation among different models, which should be treated as mutual information and not just quantifying a covariance matrix. We believe that this is a topic that deserves further investigation, and we added a paragraph in the Discussion section addressing this topic.

*3) In Sect. 4, simple averaging is used as a benchmark to compare the weighted averaging proposed by the authors. While e.g. Fig. 5 clearly show differences among the methods, it is not clear if the weighted average really provides the better (in terms of smaller disagreement from observations) estimate that the simple average. I am quite sure this will be the case, but please add and discuss the related*

*numbers.*

From the theoretical point of view, the inverse-variance weighted average is the most efficient estimator of the mean under the maximum likelihood estimation theory. The numerical results support this claim showing a much reduced variance (32% reduction) in comparison to the weighted average, and narrower prediction intervals. We added this infomration to section 4.

*My overall recommendation is to publish after these minor points have been suitably addressed.*

We hope this new version addresses well all reviewers' comments and is now suitable for publication in ESSD.

Best regards,

Carlos A. Sierra